

# The application of advanced computational algorithms used for cooperative communication transmission of vehicular networks: a proposed method

Xinyu Cui and Guifen Chen

School of Electronic & Information Engineering, Changchun University of Science and Technology, Changchun, Jilin, China

## ABSTRACT

Telematics will be one of the critical technologies in the future intelligent transportation system and establish communication between vehicles and vehicles, vehicles and networks, and vehicles and people. Thus, vehicles can sense mobile environments and make rational driving decisions. Therefore, the safety and efficiency of traffic flow would be enhanced. However, due to the unknown nature and higher complexity of the connected network environments of vehicles, the utilization of conventional optimization theory fails to generate satisfying results. To address the problem, this article proposes a methodology for collaborative transmission for communication regarding the Internet of Vehicles (IoV) with the help of advanced computational algorithms. The article employs a multi-intelligence advanced computational algorithm to construct a collaborative communication transmission mechanism in the telematics communication system model. The proposed algorithm fully considers the vehicle mobility and quality-of-service (QoS) of telematics services within the network slice. It adjusts the slice's radio resource allocation and parameter settings on an expanded time scale to improve the QoS of telematics services and increase the system's long-term revenue. The simulation results show that the proposed algorithm has a more significant performance improvement than conventional algorithms using telematics information transmission. For example, when the same load conditions are under consideration, the total capacity of the vehicle-to-infrastructure (V2I) link optimized by the proposed algorithm is still higher than that of the other three baseline strategies.

Corresponding author
Guifen Chen,
1986800019@cust.edu.cn

## INTRODUCTION

With the rapid advancement of the automotive industry and the continuous improvement of the quality of life, customers have demanded higher requirements for in-car applications since information and communication technology have also made substantial progress. In-car communication technology can provide a comfortable and safe driving environment

for in-car users and provide leisure and entertainment activities to enhance travelers' experience. As a result, in-vehicle communication has become a hot research topic for researchers and the industry globally since every achievement in this regard could make humans' lives very comfortable and convenient (*Samir et al., 2019*).

In the article, the term telematics refers to telematics technology broadly. Telematics research often considers two wireless communication technologies: short-range communication DSRC and cellular networks. It is a fact that a single wireless communication method cannot meet the needs of telematics applications (*Chen et al., 2017a*). On the other hand, the DSRC is qualified for real-time, short-range shop floor information exchange but cannot guarantee network connectivity. In contrast, cellular networks cover a wide range but have difficulty ensuring secure telematics applications with strict latency requirements. So, most of the available telematics research focuses on heterogeneous architectures based on cellular networks and DSRCs. In addition, with the development of drone technology and its popularity in civilian sectors, drones are expected to support the heterogeneous architecture of the telematics network to improve network connectivity and expand base station coverage (*Wang et al., 2017*). Although the heterogeneous architecture of telematics could support both secure and non-secure applications, the increasing demand for data traffic in the era of big data poses a challenge to the operation of the heterogeneous system. On the one hand, due to the system's limited capacity of the base stations, the quality-of-service (QoS) (*Huang et al., 2020*; *Duan, Liu & Wang, 2017*) of telematics applications cannot be guaranteed when the data volume increases. On the other hand, due to limited wireless resources, data delivery between vehicles is more prone to interference and frequent message collisions, reducing the delivery performance of secure messages, thus necessitating optimization of telematics communication in heterogeneous architectures.

Telematics has been currently divided into broad and narrow telematics. The little sense of telematics refers to information technology applied in human-vehicle interaction, such as reversing radar, reversing camera (*Tian et al., 2017*), fixed speed cruise control, road traffic sign recognition, and fatigue driving. It mainly uses Bluetooth, wired, and other technology means of communication to interact with information in a small area to help drivers during mobility. Telematics refers to information technology applied to communication between vehicles and vehicles and between vehicles and base stations in road traffic systems with broader system coverage, such as the 802.11p standard developed by the Institute of Electrical and Electronics Engineering (IEEE). Broad vehicle networking technology aims to enable information sharing between vehicles to improve road safety and reduce the incidence of accidents. In the future, car ownership will boom rapidly. How to use information technology (IT) (*Ge et al., 2017*), communication and other information technologies in road traffic rationally and efficiently has become a hot research topic for researchers.

Recently, the development of wireless communication technology has provided the theoretical basis and technical support for implementing future telematics technology. The IEEE Standards group promulgated the 802.11p standard for in-vehicle communication in July 2010 (*Su et al., 2019*), which works in the low-frequency band of 5.9 GHz (*Zhang,*

*Yang & Chen, 2018*) with a communication rate of 10 Mbps. However, for future telematics systems with massive connectivity needs, the lower communication rate, higher latency, and fewer communication nodes that could access the system could not meet future needs.

Nowadays, 5G technology has started to be commercially available on a large scale, and its advantages, such as higher bandwidth and lower latency in the millimeter wave band, have provided an improved idea to the telematics technology. Meanwhile, the 802.11ad/ay standard developed by the IEEE standards group is a directional millimeter wave standard technology working at 60 GHz. Its communication rate could reach gigabits, fully meeting the massive connection demand of future telematics (*Chen et al., 2017b*). The 60 GHz band is a free wireless local area network (WLAN) band; consumers do not need to pay when using it. This shows that millimeter wave communication technology has many natural advantages. However, as the electromagnetic waves in the 60 GHz band are in the high-frequency millimeter wave band, the disadvantages of millimeter wave are severe road loss, poorer penetration, and a smaller coverage area. To overcome these disadvantages, millimeter wave communication employs beam-focusing technology, which requires the utilization of large-scale antenna arrays. Millimeter wave terminal equipment is easy to produce and apply due to the short millimeter wave wavelength and small antennas, which can be easily integrated into communication nodes (*Jameel et al., 2018*). Through beam assignment techniques, more stable directional communication beam pairs can be formed between the system's communication nodes, enhancing the link's quality. Beam-tracking techniques could enhance the robustness of the communication link. Although beam-focusing techniques could improve the performance of high-frequency links, the enhanced version of high frequencies, when compared to low frequencies, is still not convenient for large-scale implementation in the telematics scenario due to the mobility characteristics. Still, if the high and low frequencies could collaborate, it could double the benefits to the system.

In communication systems, frames transmitted between communication nodes are generally divided into control, management, and data frames. In WIFI, management frames include Beacons, Association frames, and so on. Control frames include acknowledgment (Ack), request to send (RTS) (*Bayrakdar, 2020*), clear to send (CTS) frames, *etc*. Data frames represent data. Both management and control frames contain primarily management control information, while data frames include the media information passed between nodes. To ensure that both control and management frames are not easily lost, they are generally modulated with a low-order modulation and coding scheme (MCS) for transmission to provide a low bit error rate (BER), which makes the communication rate relatively lower. On the other hand, data frames are modulated employing a high-order MCS to transmit at the maximum possible communication rate allowed by the standard (*Niu et al., 2020*). To give full play to the respective advantages of high and low frequencies, a collaborative network architecture of high and low frequencies can be applied to the telematics system, with data frames transmitted in the high-frequency band and management and control frames transmitted in the low-frequency band, ensuring not only the reliability and stability of the connection between nodes in the system but also the efficiency of data transmission.

While telematics has a range of characteristics such as high vehicle speeds, severe wireless fading, and regularity of movement, this unique vehicle operating environment sets it apart from other wireless networks. Also, it gives it a massive challenge in terms of various key technologies (*Ghafoor et al., 2019*).

The research of adaptive collaboration modes and relay selection algorithms in telematics can effectively resolve the problem of reduced bandwidth utilization and increased latency brought about by the introduction of collaborative communication technologies, thus improving the reliability performance of the system. Therefore, a multi-intelligence advanced computational algorithm is proposed to construct a cooperative communication transmission mechanism in the telematics communication system model in the manuscript and is simulated. The outcomes suggest that it has a better collaborative communication transmission.

The rest of the article is outlined: 'Related Works' deals with the preliminary. 'The Proposed Method' presents the proposed method. 'Experimentation' is allocated to experiment and simulation with outcomes. 'Conclusion' concludes the research.

## RELATED WORKS

### Collaborative communication in vehicular networks

The basic idea of collaborative communication is based on a multiuser communication environment. A single-antenna mobile user forms a virtual multi-antenna array by sharing the antennas of multiple mobile terminals nearby, thereby gaining diversity and effectively improving the reliability of information transmission (*Dias et al., 2020*). The transmission method of collaborative communication is not a simple relay technology or the previously applied spatial diversity technology but a fusion of the two technologies. Therefore, collaborative communication technology has the advantages that both have. So, communication coverage has been expanded and shadow fading, path loss, and multipath effects have been effectively overcome. The introduction of collaborative communication technology into telematics represents a virtual multi-antenna environment where the source vehicle communicates directly with the destination vehicle. At the same time, the relay vehicle also forwards information to the source (*Zhang et al., 2018*), which undoubtedly brings diversity gains and extends the range of information transmission. However, such a high-speed in-vehicle environment leads to problems. The introduction of relaying is bound to cause more latency and reduced bandwidth utilization. Therefore, adaptive collaboration models and relay selection algorithms in vehicular networking should receive extensive attention.

The adaptive collaboration model in telematics consists of two levels of meaning. One of which is the adaptive relay collaboration policy, *i.e.,* the judgment of relay "collaboration timing" (*Fotohi, Nazemi & Aliee, 2020*), which can effectively resolve the problem of "to forward or not to forward" and "when to forward" for relay nodes. The second is the adaptiveness of the relay forwarding policy, *i.e.,* the choice of the relay node's information forwarding method when choosing a relay collaboration communication method between vehicles. Amplify-and-forward and decode-and-forward are two of the simplest yet widely

implemented fundamental forwarding strategies in collaborative communication systems. The relay selection algorithm in telematics is employed to choose the best-performing relay among all potential relay nodes when picking a relay collaboration method for communication between vehicles (*Rivoirard et al., 2018*).

In summary, the research of adaptive collaboration modes and relay selection algorithms in telematics can effectively resolve the problem of reduced bandwidth utilization and increased latency brought about by the introduction of collaborative communication technologies, thus improving the reliability performance of the system.

Vehicles in telematics can support a range of secure and non-secure applications with vehicle-to-vehicle and vehicle-to-infrastructure communications. Considering that a single wireless communication method cannot simultaneously meet the QoS requirements of multiple telematics applications. The current telematics research has focused on heterogeneous architectures of telematics (*Wang et al., 2022*). In China, Dating and Huawei have been improving and testing the V2X architecture based on LTE-V, which is based on TD-LTE technology and supports both centralized LTE-V-Cell and distributed LTE-V-Direct communication modes, which could reuse the available cellular network infrastructure without allocating dedicated spectrum, and could effectively extend the vehicle sensing range with the help of LTE-V-Direct and ensure network connectivity. The LTE-V enables ubiquitous network coverage with cellular communications, enabling the interconnection of vehicle-vehicle, vehicle-road, and vehicle infrastructure, thus effectively supporting telematics applications.

## Advanced computational algorithms in the vehicular networks

In recent years, vast amounts of data have been accumulated in various fields and the human capacity used for collecting (*Ismael et al., 2021*), storing, transmitting, and processing data has increased rapidly. These developments lead to advanced computational algorithms, which are important in enabling revolutionary advances in artificial intelligence (AI).

Advanced computational algorithms (*Bangui & Buhnova, 2021*) are classified in terms of dataset labeling: unsupervised, semi-supervised, supervised, and reinforcement learning algorithms (*Prasad et al., 2021*). Where supervised learning algorithms master a prediction function from a training dataset and estimate the output based on the new dataset. Commonly employed supervised learning algorithms include regression analysis and statistical classification methods. The training set for supervised learning algorithms consists of feature inputs and target outputs, where humans annotate the targets and all training data has annotation information. In contrast to supervised learning algorithms, unsupervised learning algorithms model the features of the training set directly and identify the underlying class rules based on the same features and targets of the training set. The difference comes from whether humans annotate the targets of the training set. Training sets of the unsupervised learning algorithms (*Islam et al., 2021*) have no human annotation information and common unsupervised learning algorithms are used for visualization, cluster analysis, and density estimation.

On the other hand, semi-supervised learning algorithms are somewhere between supervised and unsupervised learning algorithms, using partially labeled and partially

unlabeled data to master predictive functions. Finally, reinforcement learning algorithms are about achieving a goal, gradually adjusting behavior as the environment changes, and evaluating whether the feedback obtained after each action is positive or negative. Reinforcement learning algorithms are classified as instantaneous difference, dynamic programming, and Monte Carlo methods. There is still much room for research into the generalizability of those algorithms, *i.e.,* their ability to generalize needs further improvements (*Al-Shareeda et al, 2022*).

In recent years, many technologies have contributed significantly to the development of connected vehicles, where autonomous driving by computer is an ideal solution in the field of connected vehicles. Abstract autonomous driving is an advanced computational-based research, where the inputs are various types of information received by multiple onboard sensors that sense and acquire information on vehicle conditions in real-time, and the outputs are control actions such as throttle, direction, braking, and other functions. Also, due to the cyclical nature of urban traffic flows, data traffic in telematics has a regularity in the spatio-temporal dimension (*Ashish & Prakash, 2021*). When vehicles are associated with base stations of heterogeneous networks, one of the most important issues is to exploit the regularity of vehicle traffic data for load balancing between these base stations. For certain tasks, the collected data is stored and processed into the training and test sets, respectively, and the error is calculated separately. The error of the test set is then employed to represent the generalizability of the model and algorithm and the optimal strategy is again mastered to prove the algorithm's effectiveness.

When the performance of telematics is analyzed, collaborative communication technology involves algorithms of advanced computations. Middleware technology is the core technology in the current software development field, which realizes the middle program of data transmission, data filtering, format conversion, and other operations between radio frequency identification (RFID) hardware equipment and the user's application system and imports various data information read by RFID reader into telematics application through the operation of software extraction, decryption, filtering, format conversion and so on, and makes it available to telematics users through the user interface of application system (*Adnan et al., 2021*). Specifically involved are middlewares such as vehicle path navigation, emergency event handling, vehicle-assisted driving, and traffic signal control. Each one is developed regarding the standards for telematics application services. Convolutional neural networks (CNN), which are sweeping the autonomous driving technology and connected vehicle fields, employ a greedy layer-by-layer training approach to master deeply, perform feature extraction, estimate unknown functions, and utilize multiple convolutional layers to generate deeper feature maps to reach more accurate estimates of network functions. The main applications based on deep learning techniques are neural network models, sensor fusion methods, and path planning methods (*Hossain et al., 2021*). The reinforcement learning-based distributed machine-to-machine communication packet scheduling algorithm is employed in telematics communication to support million-level connectivity requirements, and the accuracy of the algorithm's predictions is evaluated using Pearson's linear correlation coefficient to measure the degree of relationship in the training set. While objective evaluation

algorithms are measured, the degree of deviation of subjective evaluation algorithms is a function of the root mean square error (RMSE). Using different advanced computational algorithms for different telematics needs to study the behavior of vehicle-vehicle (*Elgendy et al, 2021*), vehicle-roadside unit, and vehicle-pedestrian interactions in telematics and learning and exploring the overall perceptual environment of telematics. All involve the effective application of advanced computations.

The introduction of a collaborative communication system in telematics, while bringing diversity gains and improving the system's reliability, faces the problem of introducing relaying that will inevitably lead to more delays and a reduction in bandwidth utilization. In addition, the adaptive nature of the relay forwarding strategy, *i.e.,* the relay selects the forwarding method based on the channel transmission characteristics of the source-relay when a relay collaborative communication method is chosen between vehicles. Thus, amplify-and-forward and decode-and-forward are two of the simplest yet widely implemented fundamental forwarding strategies in collaborative communication systems.

The research of adaptive collaboration modes and relay selection algorithms in telematics can effectively resolve the problem of reduced bandwidth utilization and increased latency brought about by the introduction of collaborative communication technologies, thus improving the reliability performance of the system. Therefore, a multi-intelligence advanced computational algorithm is proposed to construct a joint communication transmission mechanism in the telematics communication system model in the manuscript and is simulated. The outcomes suggest that it has a better cooperative communication transmission.

## THE PROPOSED METHOD

In a real-world telematics scenario, while general-purpose mobile communication systems represent low-bandwidth, wide-coverage wireless access, wireless LANs represent high-bandwidth, narrow-coverage wireless access. Thus, a representative heterogeneous wireless network is constituted. The IEEE802.11p protocol (*Elaziz, Abualigah & Ibrahim, 2021*), currently widely employed in vehicular networking, supports short-range DSRC technology for inter-vehicle and inter-vehicle-road communication. However, the communication range of roadside unit (RSU/OBU) is generally 100 to 200 m, which is relatively short when compared to the workshop distance, making it challenging to ensure the connectivity of the communication network at higher vehicle speeds and efficiently leading to complex expansion of traffic messages. Suppose there is no other relay vehicle within the communication range of the vehicle. In that case, the vehicle must store the current data information until a relay vehicle is found that can forward the necessary information. This situation is prone to soft actor-critic (SAC) time delays. To disseminate breaking news with strict real-time requirements, the traffic messaging network must be improved to increase information reliability and real-time transmission.

### A communication model of vehicular networks

Telematics is essentially a multi-source, multi-relay, multi-destination, and multiuser network. Still, analyzing the three types of nodes concurrently when they are moving

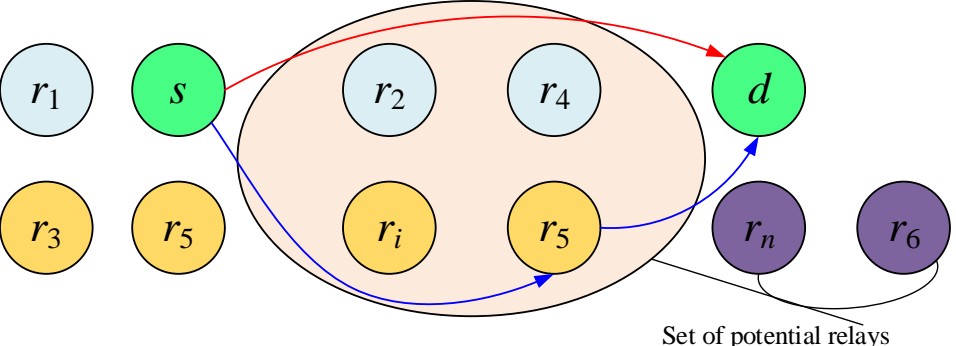

Figure 1 A communication model of vehicular networks.

simultaneously is very complex, down to the data link layer, network layer, and other protocols. Only the physical layer performance analysis is considered here to incorporate collaborative communication techniques, and the system model is shown in Fig. 1. A two-hop transmission model consisting of a source end s, several relay ends r, and a destination end d is only considered.

First, the source broadcasts a message to the destination and the relay, and the received signals $y_{s,d}$ $y_{s,r}$ for both are calculated in Eqs. (1) and (2).

$$y_{s,d} = \sqrt{P_s} h_{s,d} x + z_{s,d} \tag{1}$$
$$y_{s,r} = \sqrt{P_s} h_{s,r} x + z_{s,r} \tag{2}$$

Then, the instantaneous channel transmission characteristics of the source–destination, source-relay, and relay–destination channels are taken into account to determine how the source–destination carries out information transmission, *i.e.,* it shows when to use the collaborative communication model and when to use the direct transmission model. When the direct transmission model is employed, the relay is not involved in the communication transmission, and the instantaneous received signal-to-noise ratio of the system is calculated in Eq. (3).

$$r^d = \frac{P}{N_0} |h_{s,d}|^2. \tag{3}$$

When the collaborative communication mode is implemented, the relay end is involved in the communication transmission. Assuming that the maximum ratio merging method is used at a destination end, the instantaneously received signal-to-noise ratio of the system at this time is calculated in Eq. (4).

$$r' = \frac{P_s}{N_0} |h_{s,d}|^2 + \frac{P_s}{N_0} |h_{r,d}|^2 \tag{4}$$

## Advanced computational algorithms when multi-agent exists

The manuscript employs a multi-intelligence advanced computational algorithm to construct a collaborative communication transmission mechanism in the telematics

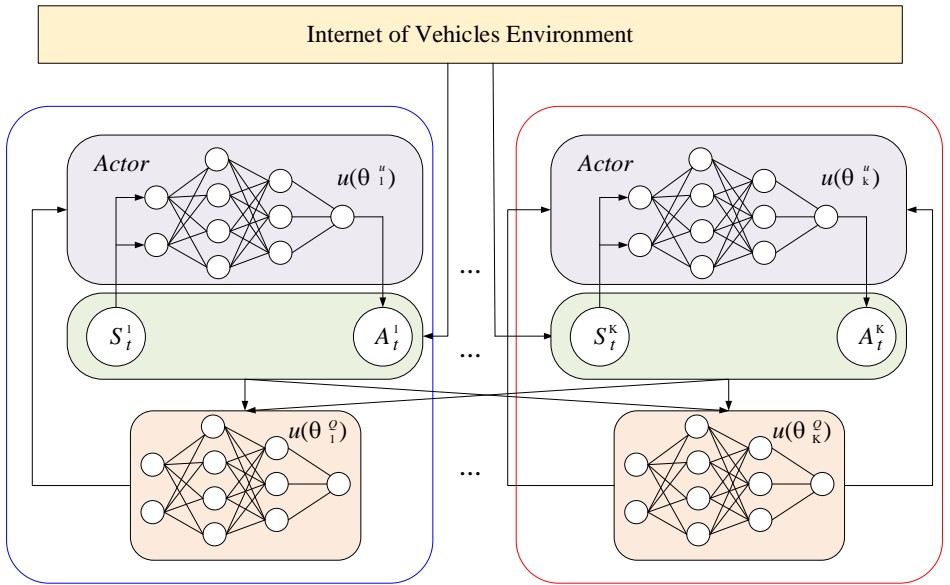

**Figure 2** **The framework of the model.**

communication system model. The framework of the model consists of K intelligence. A deep deterministic policy gradient algorithm implements each of these, and the framework is shown in Fig. 2.

The deep deterministic policy gradient (DDPG) algorithm (*Bao et al., 2021*) combines the benefits of policy gradients and DQNs. It consists of two types of neural networks: a policy-based actor-network and a value-based critic network. The actor-network collects the state of the environment and then performs the appropriate action based on the policy. The Critic network employs a state-action value function to evaluate the goodness of the action chosen by the actor-network based on the policy $Q_k(\cdot).S_t^k$ represents the input state of the intelligence $k$, $\gamma$ denotes the discount factor of the immediate reward $R_t^k$. The state-action value function of the DDPG algorithm is calculated in Eq. (5).

$$Q_k(S^k, A^k) = E[R^k + \gamma Q(S^{k'}, A^{k'})] \tag{5}$$

As a policy gradient algorithm, the main idea of DDPG is to obtain an optimal policy $\pi_k^*$ and learn the state-action value function corresponding to the optimal policy $\pi_k^*$ until convergence occurs. DDPG's Actor-Critic network employs a dual network structure. One for the evaluation network and the other for the target network are employed, where the $\theta_k^v$ and $\theta_k^Q$ of the evaluation network are updated in real time. The update process starts with a small randomly selected sample of experiences from the experience pool and feeds them into the intelligence one by one. In the training phase, the Actor and Critic networks update the parameters of the evaluation network based on small batches of samples for each input. The Critic network adjusts the parameters of the evaluation network by reducing the losses, and the loss function is calculated in Eq. (6).

$$L(\theta_k^Q) = E[(R_t^k + \gamma Q_k'(S_t^{k'}, A_t^{k'} | \theta_k^{Q'}) - Q_k(S_t^k, A_t^k | \theta_k^Q))^2] \tag{6}$$

When the Actor makes an action decision for each observation and each intelligence aims to maximize the cumulative payoff. The evaluation network parameters of the Actor are updated by maximizing the policy objective function, which is calculated in Eq. (7).

$$J(\theta_k^v) = E[Q_k(S_t^k, A^k) | A^k = \upsilon(S_t^k)] \tag{7}$$

where $u_k(\cdot)$ denotes an Actor evaluation network function for a deterministic strategy $\pi_k$ of state-mapped actions. Since the action space is continuous, $J(\theta_k^u)$ is continuously differentiable and $\theta_k^u$ can adjust the direction of gradient descent $\nabla_{\theta_k^u} J(\theta_k^u)$. As the parameters $\theta_k^u$ and $\theta_k^Q$ of the evaluation network are continuously updated, the parameters $\theta_k^{u'}$ and $\theta_k^{Q'}$ of the target network are updated using a soft update as follows:

$$\theta_k^{v'} = \tau\theta_k^u + (1-\tau)\theta_k^{u'} \tag{8}$$

$$\theta_k^{Q'} = \tau\theta_k^Q + (1-\tau)\theta_k^{Q'} \tag{9}$$

## Power distribution

When a transmitting node sends a message to a receiving node, the transmitting node generates channel contention. Only some nodes can communicate with the receiving node, at which point the other nodes are treated as interfering nodes. The power allocation in a collaborative communication system is divided into two steps: the determination of the collaborative transmitting node $i$ that communicates with the receiving node $r$ and the optimization of the power of the transmitting node $s$ and the collaborative node $i$ to increase the channel capacity of the receiving node $r$.

Here, the SINR capture effect model is implemented to select collaborating nodes. When the ratio of node signal $i$ strength to node s channel strength is greater than the threshold value $\beta$, node $r$ can receive the transmit signal from node $i$. Then, such transmit node $i$ is defined in Eq. (10).

$$\{C_r | SINR = (pCR_{i,r} GCR_{i,r})/(p_{i,r} G_{s,r}) \geq \beta\} \tag{10}$$

where $C_r$ denotes the set of collaborating nodes and $Z_r$ denotes the set of all vehicle nodes that cause interference to the receiving node $r$, except $C_r$.

In summary, the amount of interference the receiving node receives can be expressed in Eq. (11).

$$I = \sum_{j \in Z_r} \sum_{k=1, k \neq r}^{N} G_{j,r} P_{j,k} + n_0. \tag{11}$$

In a collaborative communication system, when a receiving node is close to a transmitting node, its SINR is easily satisfied by that transmitting node, and the power relationship between $p_{i,r}$ of other collaborative transmission nodes and that transmitting node can be expressed in Eq. (12).

$$P_{i,r} = P_{m,r} G_{i,r}/G_{m,r} \tag{12}$$

# EXPERIMENTATION

## Simulation environment

The telematics wireless communication scenario presented in the article helps define the telematics simulator in the city, including vehicles, lanes, and a wireless communication network model. One of the V2V links corresponds to one of the V2I links. Each intelligence's Actor and Critic networks consisted of three fully connected hidden layers containing 256, 64, and 16 neurons, respectively. The activation function is a modified linear unit, and an adaptive moment estimation optimizer is employed to train the weights of the updated neural network iteratively. The algorithm is trained for 2,000 episodes, adding a variable Gaussian noise to the actions selected by the intelligences. The probability of exploration is handled by a linear annealing algorithm, starting from 1 and annealing to 0.02 at 1,600 episodes, with the likelihood of exploration remaining constant in the subsequent training steps. In the article, a V2V link load of 1,060 megabytes is chosen for the algorithm's training phase, and a different load size is utilized when compared with other benchmark algorithms to verify the algorithm's robustness.

## Simulation results

### The convergence of the algorithm

Figure 3 shows the variation of rewards with an episode, which can observe the convergence performance of the multi-antenna deep deterministic policy gradient (MADDPG) algorithm. In the article, 100 training steps are set in each episode, and the cumulative reward sum of all training steps is utilized as the reward. The analysis shows that the reward score gradually increases as the training iterations increase. The proposed algorithm starts to converge when the training reaches 500 episodes, where the network topology changes rapidly due to the higher mobility of the vehicles. The channel fading fluctuates, leading to numerical fluctuations in the convergence of the proposed algorithm.

### The channel capacity at different nodes

Figure 4A represents the relationship between transmit node power and channel capacity for different communication schemes based on when vehicle nodes $n = 20$ are picked. The optimized channel capacity for collaborative power allocation and the allocation of cooperative communication power decrease as the power of the transmitting node increases, and the channel capacity with no collaboration keeps increasing with the power of the transmitting node. However, the channel capacity of the first two still far exceeds that of the latter. As the power of the transmitting node increases, the gap between the collaborative power allocation's channel capacity and the collaborative communication's optimized power allocation keeps decreasing. Figure 4B indicates that the channel capacity of the cooperative power allocation is slightly better than the cooperative communication power optimization allocation when the transmit node power is 5-6 dBm when the vehicle node $n = 40$ is chosen. However, when the transmit power exceeds 7 dBm, the channel optimized capacity by collaborative communication power allocation gradually exceeds that of cooperative power allocation. The channel capacity without collaboration is still much smaller than the first two. Figure 4C indicates that the communication quality

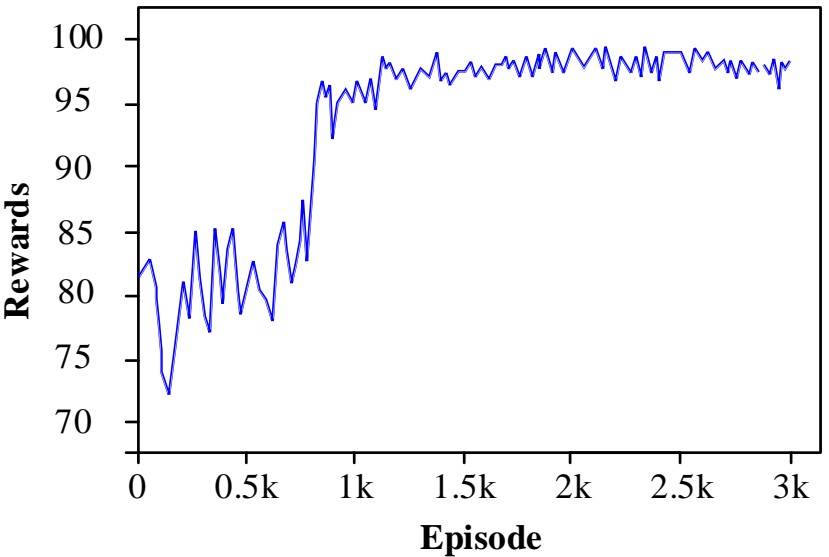

**Figure 3** **The convergence graph of the proposed algorithm.**

of the optimized allocation of collaborative communication power is consistently more significant than that of the cooperative power allocation when the vehicle node $n = 60$ is picked.

### Transmission success rates under different loads

Figure 5 shows the performance of each algorithm in optimizing V2I link capacity and performance under different V2V link payloads. The experimental results show that as the V2V link payload size increases, all algorithms' optimized V2I capacity and performance show a decreasing trend. To increase the probability of a successful V2V payload transmission, the increase in V2V payload leads to more substantial interference on the V2I link for a longer period, thus jeopardizing its capacity performance. Thus, an increase in the V2V link load will result in a longer time for the V2V link to transmit data and a higher transmit power, leading to more interference with the V2I link and consequently causing a lower capacity. Under the same load conditions, the total capacity of the V2I link optimized by the proposed algorithm is still higher than that of the other three baseline strategies. As the load increases, the total V2I link capacity of the three baseline strategies tends to decrease continuously. At the same time, the performance of the proposed algorithm shows a slight fluctuating decrease with better robustness.

Figure 6 shows the relationship between the transmission success probability of the V2V link and the V2V link transmission load size. The likelihood of the transmission success of all four optimization strategies decreases as the load increases since the amount of transmission load is fixed at the same transmission rate and the same transmission time slot, while the larger the total load size required for the V2V link is, the lower the transmission success probability of the V2V link load would be.

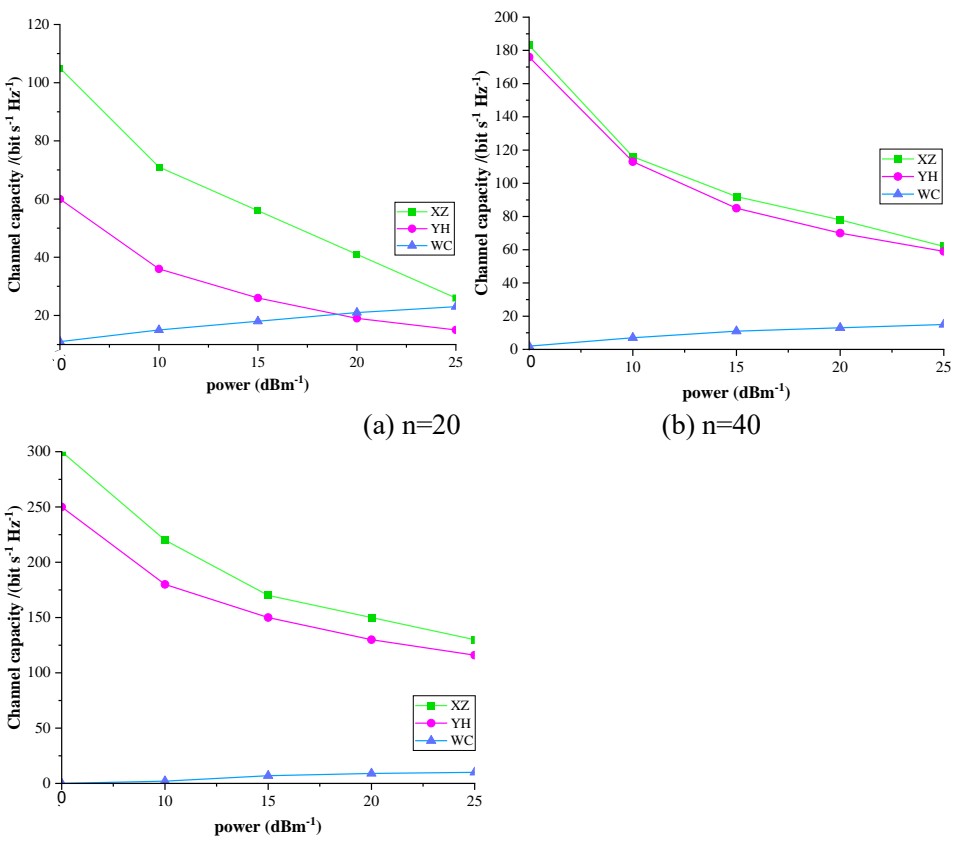

**Figure 4** **The channel capacities for different values of vehicle node n.**

### Simulation results and analysis of connectivity

To evaluate the connectivity of collaborative communication transmission in-vehicle networks with the introduction of advanced computational algorithms in the article, a simulation scenario is set up to analyze algorithms dealing with the connectivity of different mobile access point selections, in which vehicles obey a Poisson point process with a density of 0.02 m-1 on a 10 km road, and the speed follows a uniform distribution within [50, 80] km/h. The speed of each vehicle remains constant during the simulation. Assume that the communication channel obeys Rayleigh fading, the path loss index is assigned to 4, the transmit power of the mobile access point is assigned to 2W, and the vehicle receives an additive noise power of −100 dBm. In the simulation, mobile access points are selected according to different selection algorithms, and adjacent mobile access points communicate with the vehicles in between, employing a collaborative communication mode. The impact of different mobile access point selection algorithms is evaluated by comparing the received signals with the interference-to-noise ratio (SINR) and the SINR threshold to calculate the overall probability of the downlink connectivity.

Figure 7 shows the relationship between the probability of the downlink connectivity PC and the SINR's threshold $\beta$ received by the vehicle when communicated collaboratively

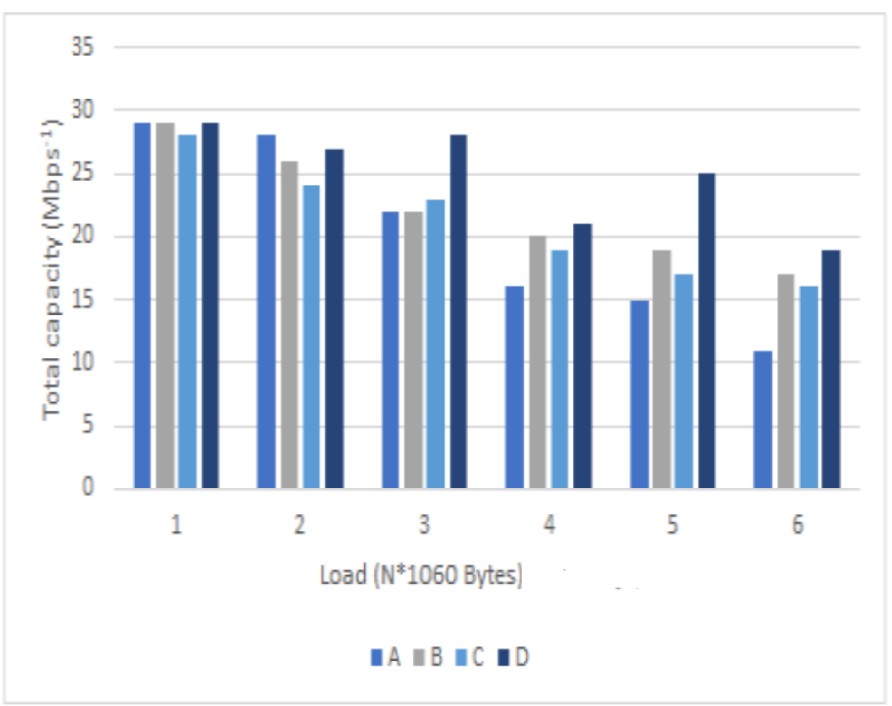

**Figure 5** **A comparison of the total link capacity under different loads.**

with two collaborating mobile access points. Three different mobile access point selection methods are compared in the simulation: independent random, sequential, and distance-based selections. The simulation results also show that the sequential and distance-based selection methods are better adapted to the actual distribution of vehicles, and their connectivity is significantly better than that of independent random selection.

## CONCLUSION

With the rapid development of intelligent transportation systems facilitating the construction and development of smart cities in which habitants travel effectively, telematics plays an increasingly important role. However, telematics is still in the early stages of its development and has many issues to be dealt with. Due to the high-speed mobility of vehicles, conventional mobile computing methods face challenges such as efficient and fast resource scheduling and power allocation. Inter-vehicle access network services are one of the most important methods for providing communication services in the vicinity of vehicles.

The research in the article is of great importance for network planning, topology control, and user experience in telematics. In addition, as a distributed wireless network, the link between terminals could be constructed without passing through the base station and transmitting data to the target terminal, reducing the relay burden of the base station.

In the article, a new collaborative communication transmission algorithm for vehicular networking is proposed using advanced computational algorithms, and connectivity is

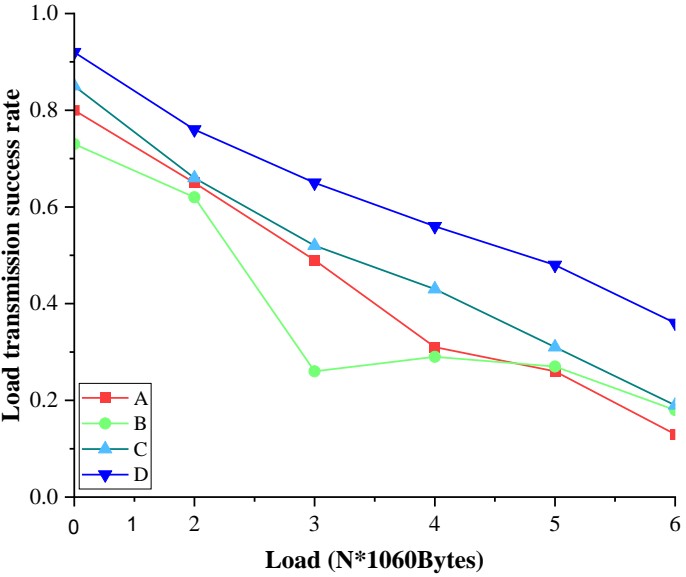

**Figure 6** The probability of successful transmission of V2V link load for different load sizes.

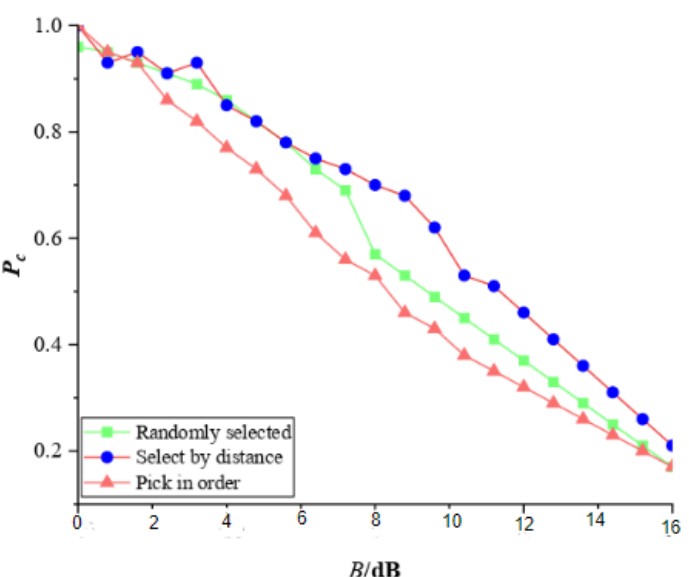

**Figure 7** The connectivity probability with different mobile access point options.

used as a fundamental but important metric for vehicular networking to evaluate the performance of the proposed algorithm. To evaluate the connectivity of collaborative communication transmission in-vehicle networks with the introduction of advanced computational algorithms, simulations are conducted.

When connectivity is a concern, the sequential and distance-based selection methods are better adapted to the actual distribution of vehicles, and their connectivity is significantly better than that of independent random selection. In addition, as the power of the transmitting node increases, the gap between the channel capacity of the collaborative power allocation and the optimized power allocation of the collaborative communication keeps decreasing. Finally, as the load increases, the total V2I link capacity of the three baseline strategies tends to decrease continuously, while the performance of the proposed algorithm shows a slow fluctuating decrease with better robustness.

### Funding
The authors received no funding for this work.

### Competing Interests
The authors declare there are no competing interests.

### Author Contributions
- Xinyu Cui conceived and designed the experiments, performed the experiments, analyzed the data, performed the computation work, prepared figures and/or tables, authored or reviewed drafts of the article, and approved the final draft.
- Guifen Chen conceived and designed the experiments, performed the experiments, analyzed the data, performed the computation work, prepared figures and/or tables, authored or reviewed drafts of the article, and approved the final draft.

### Data Availability
This is a simulation based research and did not use any external data.

We used the VANET simulator available at: https://networksimulationtools.com/vanet-simulator.

### Supplemental Information
Supplemental information for this article can be found online at http://dx.doi.org/10.7717/peerj-cs.1643#supplemental-information.

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
