# Peer review of "The application of advanced computational algorithms used for cooperative communication transmission of vehicular networks: a proposed method"

_PeerJ Computer Science, doi:10.7717/peerj-cs.1643_

## Round 0.1 · original submission · Major Revisions

Dear authors,

Two experts in the field have reviewed your manuscript, and you will see that they have could of comments for improvement of the article. Therefore, you are requested to update the article in light of those comments and resubmit for re-evaluation.

Please also improve the paper's structure and the language quality in academic writing.

**Language Note:** The Academic Editor has identified that the English language must be improved. PeerJ can provide language editing services - please contact us at copyediting@peerj.com for pricing (be sure to provide your manuscript number and title). Alternatively, you should make your own arrangements to improve the language quality and provide details in your response letter. – PeerJ Staff

Reviewer 1 ·

Basic reporting

In this work, the authors proposed a collaborative communication mechanism for the Internet of Vehicles. The proposed algorithm considers user QoS and vehicle mobility to allocate resources on the fly. The paper is well structured. Following are some of the aspects that need to be improved:

a. The title should be revised and rewritten to better reflect the research conducted in the article.
b. Abstract should be reorganized and rewritten. Also, it should contain the proposed methodology overview, the data used for analysis, and some key results derived from the conducted research.
c. Proofreading is a must. There are many places where language issues are obvious.
d. The introduction section should contain two more paragraphs: 1. How the rest of the article is outlined? 2. What is the research motivation and the contribution of the research (in bullet points)?
Moreover, the concrete research gap should be improved and clarity must prevail w.r.t to the state-of-the-art approaches.
e. Some abbreviations are used in the text. They need to be fully written before abbreviations are used.
f. the research motivation should be presented as a paragraph at the end of Section 2.
g. Instead of using the word “Equation”, authors should use the abbreviated form of “Eq.(.)”. Equations should be cited if they are not derived from the authors of the article.
h. All titles of sections and subsections should be checked and corrected to reflect better what they contain or present.
i. This sentence is extracted from the article: “The Actor and Critic networks of each intelligence consisted of three fully connected hidden layers containing 256, 64, and 16 neurons” How did the authors determine those numbers? Is there any preprocessing to reach those numbers? Please discuss it.
j. What is the MADDPG algorithm? Please clarify.
k. The authors should add some remarks to better present the meaning of Figure 7.
l. The authors should briefly mention the future agenda.

Experimental design

Comments have been mentioned above.

Validity of the findings

Comments have been mentioned above.

Additional comments

Comments have been mentioned above.

Reviewer 2 ·

Basic reporting

The article has contributed to the literature. However, it has some severe issues that need to be fixed by responding to the questions presented below.
First of all, authors should run a complete check regarding grammar, sentence structure, formal word usage, punctuation, and the presentation of the manuscript. Especially, both the abstract and introduction section should be reorganized and rewritten. Some abbreviations are just used in the text. Those should be checked and written fully where they first appear. The conclusion section should be improved. There is almost nothing found regarding the conducted research. Just some parts of other sections are repeated.

Experimental design

The article has some technical issues that need to be corrected, verified, and clarified as follows:
1. All equations should be cited properly
2. How is the simulation data generated?
3. what are the ratios of training and test sets?
4. how did the authors find the neuron numbers in each layer?
5. the contribution of the paper should be better underlined
6. What is the reward value? Please clarify it.
7. The contribution and the future direction of the research should be expressed in the conclusion section.
8. Which software is used for both simulation and running the proposed algorithm?

Validity of the findings

In continuation to the comments on experimental design section, more elaboration is required for validity of the findings.

Additional comments

No comments

---

## Round 0.2 · accepted · Accept

I am happy to let you know that the reviewers are now satisfied with the quality of the revised version of the paper. Therefore, We are happy to accept it. thank you for your fine contribution.

Reviewer 1 ·

Basic reporting

The revisions are satisfactory. However, it is recommended to fully proofread the manuscript to eliminate any language-related issues.

Experimental design

The revisions are satisfactory. No further comments.

Validity of the findings

The revisions are satisfactory. No further comments.

Additional comments

The revisions are satisfactory. No further comments.

Reviewer 2 ·

Basic reporting

Based on the revisions made and overall quality of the manuscript, I am happy to accept the article in its current form.

Experimental design

The revisions made have improved the clarity of the section.

Validity of the findings

Improved as per comments.

Additional comments

No comments.